# A Recovery-Oriented Approach: Application of Metacognitive Reflection and Insight Therapy (MERIT) for Youth with Clinical High Risk (CHR) for Psychosis

**DOI:** 10.3390/bs14040325

**Published:** 2024-04-15

**Authors:** Bethany L. Leonhardt, Andrew C. Visco, Jay A. Hamm, Jenifer L. Vohs

**Affiliations:** 1Department of Psychiatry, Indiana University School of Medicine, Indianapolis, IN 46202, USA; avisco@iupui.edu (A.C.V.); jvohs@iupui.edu (J.L.V.); 2Sandra Eskenazi Mental Health Center, Indianapolis, IN 46202, USA; jay.a.hamm@gmail.com; 3College of Pharmacy Practice, Purdue University, West Lafayette, IN 47907, USA

**Keywords:** clinical high risk, metacognition, recovery, psychosis, psychotherapy

## Abstract

Clinical High Risk for psychosis (CHR) refers to a phase of heightened risk for developing overt psychosis. CHR often emerges during adolescence or early adulthood. CHR has been identified as a group to target for intervention, with the hope that early intervention can both stave off prolonged suffering and intervene before mental health challenges become part of an individual’s identity. However, there are few treatment modalities that can address some of the specific needs of CHR. Metacognitive Reflection and Insight Therapy (MERIT) is an integrative psychotherapy that can be applied to the CHR population. MERIT offers unique advantages to working with the CHR population as it aims to improve self-direction and recovery through stimulation of metacognitive capacity, a phenomenon that has been associated with recovery. This paper explores unique aspects of the CHR population and how MERIT can address barriers to recovery for individuals experiencing psychosis-like symptoms. Several case examples and a clinical vignette using MERIT to support patients with CHR are offered to exemplify this approach. MERIT offers a way to assist persons with CHR to address barriers to their personal recovery and to develop nuanced understandings of ways to master challenges.

## 1. Introduction

In recent years, there has been increased interest in intervening early in the course of serious mental illnesses, such as psychosis. Many have emphasized the importance of intervening before psychosis is fully present, and thus have argued for treatment for individuals at risk of developing psychosis (referred to as individuals with Clinical High Risk (CHR) for psychosis) [1]. Individuals meet the criteria for CHR for psychosis if they have the presence of psychotic-like symptoms and are help-seeking [2,3]. Individuals with CHR differ from individuals with psychosis in terms of their conviction regarding psychotic-like experiences. For example, instead of being convinced that they hear someone else’s voice saying derogatory things, an individual with CHR may attribute these voices to their own mind [4]. The prevalence of CHR is still being investigated [5], but recent research has reported rates of 19% in clinical samples and a much lower rate of 1.7% in the general population [6]. While it is uncertain which individuals experiencing CHR will go on to experience psychosis, research has shown that CHR individuals experience functional impairments whether they experience a full psychotic episode or not [7].

The aim of intervening at such an early point in the potential onset of psychosis is to stave off and prevent the additional suffering that can accompany a full psychotic episode and to assist persons with CHR to be able to live a full and meaningful life. As this is an emerging area of study and intervention, there is no gold standard for which type of treatment is most effective in preventing increased suffering for those with CHR, but it is recommended that psychosocial interventions are the primary treatment and that psychotropic medication be used sparingly [2,8]. Psychosocial interventions, such as individual therapy, are emphasized to assist persons with CHR to make sense of the experiences they are having and to find ways to manage their distress so that it does not compound and develop into a more serious condition [9]. A recent systematic review [10] noted that most treatments for CHR include an emphasis on psychosocial programming, including family interventions, cognitive behavioral interventions, motivational interviewing, substance misuse interventions, and skills training. One therapy modality that may be particularly well suited to assist persons with CHR to make sense of their experiences and find ways to master them is Metacognitive Reflection and Insight Therapy (MERIT) [11]. In this paper, we offer general rationale for how MERIT might be an ideal treatment for the CHR population, and then we explore specific considerations of each element of MERIT for working with individuals experiencing CHR based on our experience with implementing MERIT in a CHR clinic. We illustrate how the approach can be applied to those with CHR through a case vignette. Finally, we discuss some key findings from using MERIT with individuals with CHR.

## 2. MERIT and CHR

MERIT is an integrative psychotherapy framework specifically designed to address deficits in one’s ability to think about oneself and others, and to then use this knowledge (referred to as metacognition) to respond to the problems one faces in life. Metacognition includes a spectrum of activities ranging from discrete to synthetic and comprises the following four domains: self-reflectivity (understanding one’s own mind), understanding the mind of the other, decentration (ability to see oneself as part of a larger whole), and psychological mastery (being able to apply reflection about self and others to respond to the challenges one faces in life) [12]. An example of discrete metacognition would include the ability to identify different cognitive operations in one’s own mind, such as knowing that one is having a memory or a desire. Synthetic metacognition involves integration of different aspects of oneself, such as understanding how an event earlier in one’s life may create a specific emotional state later in life. Individuals experiencing psychotic disorders have been shown to have deficits in metacognition (see [13] for a full review) in both the early and later phases of illness [14] and at a more severe rate than other mental health conditions [15]. Deficits in metacognition are connected to a range of poorer outcomes, including increased negative symptoms, poorer work performance, decreased intrinsic motivation, and impaired self-recovery [15,16,17]. In light of these connections to functioning and outcomes, metacognition is an important target for treatment.

MERIT is rooted in the idea of personal recovery, seeing the patient as an active agent trying to make sense of their challenges and helping them to move toward living a life that is personally meaningful to them [11]. MERIT has a practice framework that consists of eight elements to assist persons in developing their metacognitive capacity and move toward recovery. Empirical evidence for MERIT’s effectiveness has included two open trials of MERIT for individuals experiencing psychosis; both trials showed improved metacognition, high rates of acceptability, and no adverse effects [18,19]. Randomized controlled trials have also shown good outcomes, including high levels of feasibility and acceptance in real world settings, improved metacognition, improved insight, and no adverse effects [20,21,22]. Additional evidence for MERIT’s effectiveness and acceptability comes from a myriad of case studies examining MERIT with individuals with a range of presenting problems (see [13] for a summary of case studies). Case studies do not provide the same level of evidence as more systematic trials but present in-depth rich accounts of how MERIT has helped unique individuals to move toward personal recovery. Several case studies, and one trial that focused on MERIT’s potential to promote insight, have been in FEP clinics [22,23,24]. Vohs and colleagues [22] reported that individuals experiencing FEP who received 6 months of MERIT demonstrated clinically and statistically significant improvement in insight when compared with FEP participants who received supportive therapy in the control group. Two case studies by Leonhardt and colleagues [23,24] reported in-depth analysis of how MERIT helped two individuals experiencing FEP to achieve meaningful gains in their metacognition and personal recovery. MERIT’s effectiveness with young adults with recent onset psychosis offers a rationale for attempting to reap similar benefits even earlier on in the progression of problems in young people. Additionally, there appear to be at least two ways in which MERIT’s practice framework is uniquely relevant for this population.

First, MERIT is well suited to assist persons with CHR to move toward recovery due to its flexibility to be applied to people across the spectrum of metacognitive abilities. MERIT’s assessment and intervention framework operationalizes metacognition through hierarchical scales, encouraging therapists to adjust their interventions to match the individual’s specific level of metacognitive abilities. Thus, individuals with varying levels of metacognitive capacity can be effectively treated with MERIT, as the therapist is assessing the patient’s metacognition at each session and offering interventions at that level and helping them to move to the next level of metacognition, referred to as scaffolding, where the therapist helps the patient understand how to integrate what is happening in their mind in a slightly more complex way. For example, if a patient is at a level of metacognition where they could recognize and differentiate their cognitive operations (such as desires being distinct from memories) but they struggle to identify a range of nuanced emotion (the next highest level of metacognition), the therapist would offer interventions to help explore emotions and help differentiate between a range of emotions so that the patient could name their own emotional experiences. This has particular advantages for individuals experiencing CHR as they tend to be younger, and youth have shown to have lower levels of metacognition, even in healthy controls [25,26]. It has been hypothesized that there is a developmental aspect to metacognition and, as youth move through their own development, they are able to see themselves and others in increasingly complex ways and thus have more strategies available to them to respond to the challenges that arise. For youth experiencing CHR, they are likely to have challenges in metacognitive capacity related to development, and challenges in metacognitive capacity related to the experience of psychotic-like symptoms. Thus, youth in general may benefit from a metacognitive approach that is attuned to the level of metacognition they currently possess, helping them develop more complex levels of metacognition. The flexibility of MERIT enables therapists to intervene even with individuals with severe metacognitive deficits, helping these patients with CHR to develop more complex and integrated metacognitive capacity that they can then use to manage the challenges of young adulthood.

A second advantage to using MERIT with a CHR population is MERIT’s focus on recovery and promoting self-direction. Given that schizophrenia typically has onset in early adulthood, individuals identified as CHR are often adolescents or young adults [27]. Individuals in this age range tend to be wrestling with ideas about their future and developing their own autonomy, thus MERIT could offer support and reflection around these dilemmas. Rather than a more prescriptive or skills-based approach, the open and reflective nature of MERIT could assist persons with CHR in identifying what matters to them and what a meaningful life looks like from their unique perspective. This is consistent with calls for treatment to support personal recovery [28,29] and with data suggesting those with CHR have more negative self-esteem and less cohesive sense of self [30]. This can also aid in combating or diffusing self-stigmatizing beliefs, as MERIT can help persons to have a nuanced view of themselves rather than accepting an illness label. Next, we discuss the elements of MERIT in more detail and explore specific considerations for applying MERIT to a CHR population.

## 3. MERIT Elements for CHR

MERIT consists of eight core elements that must be present in each session. Sessions are unstructured and MERIT has an integrative flexible framework from which therapists from many backgrounds can implement its elements in the flow of conversation. MERIT should be conducted after the patient has been able to give their informed consent, and standard measures should be taken to ensure ethical practice, including safeguarding confidentiality. As MERIT elements are described in detail elsewhere (e.g., [11]), we will present here a brief description of each element followed by an exploration of what that element may look like working with a CHR population. Reflection on applying each element to the CHR population comes from work conducted within a CHR clinic in the Midwest. The CHR clinic is part of a larger community mental health center and aims to provide preventative care to individuals identified as at risk of developing psychosis through the use of the Structured Interview for Psychosis-Risk Syndromes (SIPS; [31]). Additional criteria include an age range of 14–35 years old, IQ above 70, and no previous history of a primary psychotic or substance use disorder. The authors on this paper served as therapists (ACV, JLV) and supervisors (BLL, JLV) of the clinical work conducted in this CHR clinic.

## 4. Element One: The Agenda

The first element of MERIT is the agenda, where the therapist attempts to understand what the patient is seeking from the session. This differs from other modalities (e.g., CBT), which may set an agenda for the session similar to a business meeting, and refers instead to the therapist exploring what the patient desires, consciously or unconsciously, through their words and actions in the session. For example, a patient may present with an agenda to be understood, or to be supported and agreed with, or to entertain the therapist [32]. Patients may have multiple agendas or the agenda may change throughout the session. In a MERIT session, the therapist views patient behavior and speech as purposeful, and seeks to reflect on these purposes with the patient.

For individuals experiencing CHR, this is often their first encounter with the mental health system as they tend to be adolescents or young adults [27]. Thus, patients with CHR often present to therapy wanting advice or direction. In our experience, youth with CHR often want the therapist to tell them what to do or view the therapist in a parental role, expecting that the therapist will have an answer to the problem that they are discussing. When the therapist might name this (“You want me to tell you how to solve this problem”), patients often agree and feel confused at the possibility that this would not be the case. These conversations often lead to opportunities to reflect with patients about parental figures in their life and support autonomy and self-direction in being able to reflect on what the patient wants in that moment rather than relying on an adult to give them the answer.

Another common theme in addressing youth with CHR is that they are often not sure why they are in therapy or what they can talk to the therapist about. It is a common experience for many of the youth in our clinic that their parents have sent them to treatment, and the youth experience some confusion around what they are supposed to be doing or how they are to benefit from therapy. Addressing these themes in session helps patients begin to formulate an idea about what has transpired in their lives that has brought them into the mental health system, and helps patients to reflect on how exploring their experiences with another person could help them to better understand themselves, others, and to then formulate strategies to respond to the problems they face in life.

## 5. Element Two: Insertion of the Therapist’s Mind

The second element of MERIT is insertion of the therapist’s mind, referring to the recognition of the importance of psychotherapy as dialogical. In MERIT, therapists are seen as consultants, not to give advice or solve problems for patients, but as a thinking person in the room to explore topics together. The therapist discloses their own ideas to foster and sustain intersubjectivity [33], to scaffold an understanding of the mind of other people, and to prompt reflection from the patient about the therapist’s thoughts and reactions. The therapist inserts their thoughts without the assertion that their thoughts are somehow more “correct” than others, instead offering their thoughts as a way to stimulate ongoing shared reflection about the patient.

Due to the dynamics explored in the previous section regarding agenda with patients experiencing CHR, therapists utilizing MERIT with this population often have to balance inserting their mind without assuming a parental or educational role. For example, it is common for patients in our clinic to share stigmatizing views about themselves for having experienced psychotic-like experiences. In such instances, it seems important for the therapist to share their thoughts about stigma and, at times, appropriate to provide information regarding recovery and positive outcome possibilities for individuals who experience psychosis or psychotic-like experiences. However, this could easily reinforce the patient seeing the therapist as an authority figure with years of experience and a degree, and thus make it more challenging to mutually explore therapist and patient thoughts without privileging the therapist’s thoughts as more “correct”. To address these concerns, therapists in our clinic ask whether it is helpful to share information or therapist thoughts, and then invite reflection from the patient about the therapist’s thoughts that are shared. Therapists are attentive to the dynamic of being placed in (or assuming) an authoritative role and name these dynamics when they occur, continuing to voice support for the patient making their own sense of what is unfolding and encouraging them to be an equal participant.

## 6. Element Three: Eliciting the Narrative Episode

The third element of MERIT is eliciting narrative episodes, which includes reflecting with the patient on specific moments in their life and exploring the internal states of the patient and others within those narratives. Narrative episodes are essential to build the capacity for the individual to better understand themselves and others, and to start to form ideas about how events in their life have impacted them over time, thus forming a more integrated and nuanced sense of oneself and others. To achieve this, MERIT therapists ask about specific moments where a patient felt or thought a certain way and attempt to place these events within the context of the patient’s life.

This element can be particularly challenging with the CHR population, as being young often means a limited amount of life experience for them to reflect upon, and youth are less likely to reflect upon their lives as a function of being young [34]. Often, when attempting to elicit narrative episodes, patients state that things have always been this way and struggle to reflect on a specific instance or to note when something started in their life. To assist in building this capacity, the therapist finds it helpful to offer narratives of their own life to help demonstrate how to reflect on a moment in one’s life. Patients often find this helpful, especially when the tone of the narrative matches the experience they are trying to explore, as they are able to relate to the therapist’s experience and then explore how it is similar or different from their own.

An additional way that the lack of narrative episodes may be addressed with individuals with CHR is to explore entertainment content in which the youth is interested. For example, knowing which anime a patient is interested in or movies they enjoy may offer a way for the therapist to explore with the patient what is compelling about these stories, which characters they identify with (or which ones they do not like), and may then invite reflection about how this matches or differs from the patient’s life. For example, one patient described to her therapist that she enjoyed playing a tabletop role-playing game with her family and a small group of friends. This patient often found it challenging to share narrative details of her life but was able to describe the development of her game characters in great detail. Through these descriptions, the therapist was able to notice with the patient how some of the characteristics seemed to align with the patient’s thoughts about herself. The patient then shared how some of the characters possessed characteristics that she wished she possessed herself. The therapist and patient were then able to use these moments to think more about experiences from the patient’s own life, her capabilities, and her interactions with others.

## 7. Element Four: Defining the Psychological Problem

The fourth element of MERIT is defining the psychological problem, which refers to naming and exploring with the patient what it is they struggle with. The psychological problem is not reducible to only a diagnosis or symptom of mental illness, rather it must be something that the patient identifies as a problem in their life. This could include being lonely, struggling to feel motivated, or difficulty understanding other people and thus not being able to form relationships that feel adaptive. In other words, in MERIT, the focus of treatment is not necessarily on symptoms of a mental illness but on what the patient states they want to change about their life. This can include symptoms of mental illness but often includes other concerns.

Patients with CHR tend to present to therapy without being able to name a clear psychological problem. Rather, there is a general sense that things are not going well or that other people think the patient needs help. Often, patients are unclear about exactly what is driving their distress. Patients in our clinic describe not knowing how they are feeling or are confused about why they feel a certain emotion. This is an essential place to focus therapy, attempting to name and explore internal states and then exploring with the patient the narrative events that may have preceded the distress. For example, one patient described to his therapist a distressing experience when he threw a water bottle across a room while with his family. He stated he was crying and upset when he threw the bottle, but struggled initially to describe what it was that made him upset or to name emotions in a nuanced way, reporting only that he was upset. Through interventions focused on scaffolding metacognition, the therapist was able to help the patient name the feelings in his body, eventually labeling them as anger. The therapist then helped the patient to reflect on the events of the day, noticing where he was, who he was with, and trying to notice how the patient’s internal states changed throughout the course of the day. Eventually, the therapist and patient came to identify that the patient was feeling jealous of a sibling who was receiving attention the patient desired, and that the patient tended to push away such feelings until they felt unmanageable, thus becoming a psychological problem, which the therapist and patient continued to explore.

Patients also struggle to identify psychological problems within the context of their family unit, often internalizing familial dysfunction. Several patients in our clinic have presented for assessment and treatment with their parents and, over the course of treatment, it became clear to the treatment team that complex family dynamics were impacting the patients. Stimulating decentration in these cases is a critical task and a challenging one. For example, a patient presented for services with his family, and the patient’s parents dominated the intake process, speaking for their child, interrupting one another, and bringing up their own mental health concerns throughout the interview. While these parents were clearly well meaning and help-seeking, their expressed emotion elicited anxiety in the patient. Over the course of treatment, the patient identified that they believed they were the cause of their family’s dysfunction and the marital discord between their parents, and they expressed feeling guilty that they were taking up valuable resources like time and money. The therapist was able to scaffold narratives by sharing his own experience with family dysfunction and link it to the patient’s expressed psychological problem to stimulate decentration. The patient was then able to consider alternative possibilities, e.g., “that adults fight and have problems and that might not have anything to do with me; perhaps I am feeling bad because there is so much going on in the house and it is stressful”.

## 8. Element Five: The Therapeutic Relationship

The fifth element of MERIT concerns attending to the therapeutic relationship by exploring with the patient what it is like to meet with and discuss their life with the therapist. This is a foundational part of many therapeutic orientations and, in MERIT, it is viewed as an essential way to explore interpersonal processes. Metacognitive acts do not occur in a vacuum but rather occur and gain meaning from our interactions with others. As such, the therapeutic relationship is a key vehicle for the patient to start to explore what it is like to encounter another person and to reflect jointly with the therapist about that experience and how that helps the patient to form ideas about themselves and others.

For individuals with CHR, they often think of the therapist as the authority figure and thus it is a unique experience to be invited to reflect on that with the therapist. Patients often note that they do not talk openly with other adults in the same manner that they do with the therapist, and this is frequently their first opportunity to reflect on the usefulness of a relationship with an adult. Patients are curious about the therapist’s experience, wanting to know if the therapist is able relate to their experiences and how the therapist resolves similar dilemmas in their life. The novelty of these types of conversations appears useful to patients; they describe that relating in this way to an adult helps them to feel more in charge of their own life. For example, a patient expressed that meeting with the therapist helped him to feel more confident in communicating his needs with his family. Prior to therapy, he often felt like he could only go to his parents for practical solutions. As his self-reflectivity increased, he began to see himself and his psychological problems in a more nuanced way and would experience frustration when his family would offer seemingly simple answers to what he felt were complex problems. Eventually, he realized that he sometimes found it helpful when they offered practical advice/answers but, in most cases, he actually wanted to feel “listened to” and to feel like he could reflect with his parents about situations and come up with his own answers within their supportive environment, similar to what he experienced in therapy. Instances such as these may aid in additional reflection regarding the therapeutic relationship, as the therapist and patient may identify the novelty of therapy and the therapist themself, while also thinking about how helpful aspects of the therapeutic relationship could possibly be mirrored by the natural supportive relationships present in the patient’s life.

## 9. Element Six: Reflecting on Progress

The sixth element of MERIT concerns reflecting on the progress of therapy, both within the session at hand and during therapy overall. Within the session, this can refer to how the patient’s thinking or emotional state changes because of the conversation with the therapist. Within the larger course of psychotherapy, this can refer to how the patient is able to apply the reflections from therapy to the problems that arise within their life.

For individuals with CHR, this element could be particularly challenging, as youth tend to have difficulty assessing the temporary nature of their emotional states, often describing that they have been feeling negatively “forever” when, upon further reflection, it may be clear they have been feeling that way for a few months. Relatedly, patients often express frustration when they do not immediately feel better following a difficult session, describing an expectation that things should improve rapidly since they are seeking treatment. In such interactions, the therapist shares their thoughts about the length of time it can take to notice change and to normalize difficult feelings and situations that may not be resolving immediately. In exploring the patient’s and the therapist’s perceptions of progress, the therapist is able to help increase the patient’s awareness of change that is occurring, even if it is hard to detect due to the incremental nature of change, and help increase the patient’s sense of buy-in to the process of change. For example, one patient expressed frustration that it felt like she would never entirely be rid of her mental illness and that if she continued to experience symptoms that it meant she was a failure. The therapist may use these opportunities to stimulate self-reflection and reflect on progress. Perhaps it is true that she would never feel a complete absence of symptoms, but does this mean that she is doomed to a life of suffering? The therapist could offer narratives from the patient’s life that instead are strong examples of the resilience that she, the patient, has been building, and notice with the patient how her understanding of herself and her psychological problems have changed from when she first began therapy to the present. The patient identified that, while she still felt frustrated by the presence of symptoms, she could notice specific examples of how she handled things differently, improved her relationships with others, and felt less overwhelmed by life at large. Identifying this process specifically as resilience was helpful to her in the sense that she did not need to have all the answers or experience perfect happiness, but instead realizing that she wanted to feel confident that she could understand and manage her emotions in the face of life and herself becoming more complex.

## 10. Element Seven: Stimulating Self-Reflection and Awareness of the Other

The seventh element of MERIT concerns correctly assessing the patient’s level of metacognition regarding their self-reflectivity and awareness of the internal states of others, and then providing interventions at that level of metacognition or scaffolding to the next highest level of metacognition. As persons develop improved metacognitive capacity, they are able to integrate more ideas into increasingly complex and nuanced understandings of self and others. It is essential that therapists correctly assess the patient’s metacognitive capacity so that they offer interventions that are accessible to the patient. For example, if a patient does not recognize that their thoughts are fallible and can change over time, offering interventions to challenge a thought or reflect on the subjectivity of a thought are not going to be effective. Similarly, if a patient is struggling to identify emotions within themselves, interventions that invite the patient to reflect upon their emotional state are likely to fall flat. Thus, the MERIT therapist must assess a patient’s level of metacognition at each session and tailor their interventions to match the level that the patient can access and tailor scaffolding to the next level of metacognition.

For individuals with CHR, a common theme with this element is patients exhibiting very low levels of awareness of others’ minds. This is likely due to age and developmental stage, as studies have shown that younger individuals tend to score lower on awareness of others even if they do not experience a mental health concern [26]. Given the tendency for younger individuals to have less complex awareness of the internal states of others and the likelihood that CHR patients are living with family members and reporting dysfunction in those relationships, it is an imperative part of psychotherapy to assist patients in gaining capacity to reflect on what others may be thinking or feeling. Frequently, patients may describe how they feel about someone else in their life, such as knowing that their father makes them angry, but they struggle to think about their father’s motivations or details of their father’s life that may influence their father. This element requires quite a bit of scaffolding from the therapist to assist patients in thinking about the internal states of others and helping them, often for the first time, to mentalize about significant others in their lives.

## 11. Element Eight: Stimulating Mastery

The eighth and final element of MERIT concerns assessing and stimulating the correct level of psychological mastery. Mastery refers to using knowledge one has of oneself and others to respond to the problems that arise in life. Consistent with the previous element, MERIT therapists aim to intervene at the appropriate level of mastery that the patient can access and to help scaffold them to the next highest level. The mastery scale starts with the definition of a clear psychological problem and then increases in complexity regarding how the person responds to their problem. For example, a less metacognitively complex response to a problem would be actively avoiding something that makes one anxious, whereas a more metacognitively complex response would be to understand why something makes one anxious and then using the understanding of why to combat anxious thoughts.

For patients experiencing CHR, a common issue the therapist encounters when trying to stimulate mastery is the challenge of promoting the patient’s sense of agency in the face of often limited autonomy. This is often due to the age of the patients; as youth, they often perceive that they are not in control of their lives. While it may be true that their autonomy is limited in some ways, it is also true that they remain agents in their own lives, making decisions about how to manage their internal states. Accordingly, the therapist offers interventions to help patients to see the nuance and distinction in this dilemma.

Another common consideration for stimulating mastery with the CHR population is that their family members often must be included in these conversations. A frequent dynamic that emerges in our work with CHR patients is that, when the patient starts to experience a more intense emotion that they are working on mastery strategies for, the patient’s parents may become alarmed, likely due to the parents’ own troubling experience of their youth’s struggles with mental health and concern that this indicates something serious. Emotion regulation and risk tolerance are two essential topics of conversation with both youth experiencing CHR and their family supports to help families better understand the delineation between distressing but manageable emotions and an acute psychiatric crisis. These conversations also include addressing stigmatizing beliefs about psychosis and what it means to be considered CHR, as family members often worry that, because their youth has this CHR label, they are doomed to experience psychosis. Also common are beliefs that experiencing psychosis is equivalent to a prognosis of lifelong disability, dependence, and suffering. 

A final consideration for stimulating mastery with a CHR population is that, due to their limited life experience, CHR patients tend to present with very low levels of mastery. As such, the therapist must provide more psychoeducation about ways to manage distressing experiences than is often seen in MERIT with older populations who may have more experience of trying to manage their distress and thus have a better sense of which approaches work or do not work for them. Similar to previous comments, this requires the therapist to strike a balance between offering their expertise and creating a non-hierarchical relationship in which patient and therapist are reflecting jointly.

## 12. Clinical Vignette of MERIT with CHR Patient

Following, we offer a case illustration of long-term MERIT with an individual diagnosed with CHR. We highlight how each of the MERIT elements impacted the case conceptualization and positively influenced progress. The patient provided verbal consent to use his clinical material and all personally identifying information has been disguised to protect his confidentiality.

Presenting Problem and Patient Background

Ryan was a white male in his late teens who presented for assessment after being referred to the clinic by a private practitioner after that practitioner had concerns related to psychosis during their initial appointment. Ryan was treatment naïve at the time of screening and had only ever received supportive services from guidance counselors at school. He was raised by his father and mother in a middle-class suburban neighborhood in a Midwestern city with his two siblings. Ryan and his family reported that he had never been separated from his family and that his mom and dad were in a committed relationship. Ryan’s mother reported no complications during her pregnancy nor during labor and delivery. Ryan was described as a healthy child throughout development, and it was only during his teenage years that he had any health concerns. When he was 15, he was hospitalized after his appendix burst and he was treated for an internal infection. When asked about this situation, Ryan described it as being traumatic. He felt like he was going to die and it was challenging to be separated from his family for a prolonged period.

Ryan described growing up as “pretty normal”; he engaged in activities outside of the home such as organized sports; he remembers always having a robust group of friends. He routinely engaged in activities with his family like going to church, having family game nights, and visiting with extended family. He was a good student and his mother described him as being a high achiever. He shared that his parents would fight from time to time but, overall, it seemed like they had a healthy relationship. Ryan described feeling close to his mom and he appreciated that she was reliable and caring. He shared that he would like to be closer to his dad but denied having any relationship issues with his father. Ryan’s mother did not endorse any personal history of mental illness. She disclosed that Ryan’s father had been diagnosed with major depression and had struggled with suicidal ideation off and on for several years. She shared that Ryan’s father was not in treatment at the time of Ryan’s intake but did not express any overt concerns about him.

When thinking about his own mental health, Ryan shared that the only time he remembered a shift in his mental health was around 12 years old. He noticed that at this age it felt like something “changed in me”. He began to feel more anxious, worried that bad things were going to happen to him and his family. He also began to feel insecure in his friendships and was worried that he might hurt other people in some way, despite having never done anything to harm others previously. Ryan expressed that this feeling would wax and wane over the years but that it started to worsen leading up to his seeking services in the CHR clinic. He noted that he began to have a more and more challenging time at school, particularly during the COVID-19 public health emergency, and that engaging in school remotely was not a good experience. He shared that being separated from his friend group was really challenging, particularly being separated from a group of friends with whom he consistently played a fantasy role-playing game.

During this time, Ryan noticed that he would occasionally hear an antagonistic voice. He made sense of the voice by believing it to be his own thoughts being spoken aloud. He denied that the voice was external to him and did not demonstrate delusional conviction when attempting to account for the experience. He endorses being distressed and distracted by the presence of the voice and began to be concerned about what this could mean about him as a person or what the future may have in store for him. Ryan experienced internally stigmatizing thoughts and concerns, and guilt, noting that he felt he was being punished. Additionally, Ryan endorsed an increased sensitivity to sound and light. He also endorsed other symptoms that seemed consistent with anxiety and depression. Ryan denied any issue with suicidal ideation at the time of assessment but endorsed a history of suicidal ideation with no attempts or plans. He noticed that his suicidal thoughts were often exacerbated by thoughts that he was worthless, defective, and “a pretend person”.

Start of Treatment

At the outset of therapy, Ryan had been diagnosed with Attenuated Positive Symptom Syndrome (APSS) and social anxiety. He was very polite, well put together, and expressed that he wanted to “do well” in therapy. Being 16 years old and the first-time receiving treatment in a clinical setting with a treatment provider, he was often accompanied to appointments by his mother and would sometimes request that she sit in on therapy appointments for the first several minutes. During these initial sessions, he would seemingly defer to his mother to help express what was bothering him or ask her to relay something that had happened at school or at home. During these sessions, there would be a matter-of-fact recounting of events and Ryan would demonstrate feelings of distress but had a challenging time articulating his experience on his own. Ryan had a notable perceived lack of agency during this time and seemed confused about “how therapy works”, being uncertain about what he could talk about and deferring to the therapist when starting sessions.

At the beginning of psychotherapy, Ryan’s metacognitive abilities were assessed as impaired using the Metacognition Assessment Scale-Abbreviated (MAS-A; [12]). MAS-A consists of the following four scales thought to measure salient aspects of metacognition: self-reflectivity, understand the mind of the other, decentration, and mastery. Each subscale ranges from discrete metacognitive activity to more synthetic integrated metacognition. As such, lower scores reflect less complex metacognition and higher scores reflect more complex metacognition.

In terms of self-reflective capacities, Ryan could identify mental activities occurring in his own mind and distinguish between the various cognitive operations such as memories, desires, thoughts, etc. However, he was unable to identify nuanced states of emotion, typically describing events as either “good” or “bad” or utilizing more one-dimensional emotion words such as “happy” or “sad”; he was accordingly unable to engage in higher levels of self-reflection such as linking various events over time to form and reflect upon more complex ideas of himself. Similarly, Ryan was aware that others had mental activities of their own but was unable to differentiate between the range of internal states that others might have, such as their own thoughts, wishes, or intentions. In terms of decentration, Ryan generally perceived himself as the center of all activities, was unable to see events from multiple perspectives, and did not have a clear understanding of other people having lives entirely separate from him. Finally, in terms of metacognitive mastery, Ryan often struggled to articulate a plausible psychological problem. He tended to resort to discussing only positive surface-level parts of his life and only sometimes could express a general sense of distress without much ability to elaborate on this. Taken together, his metacognition on the MAS-A scale at the beginning of the course of therapy was as follows: self-reflectivity (3), understanding mind of the other (2), decentration (0), and mastery (1.5). We review the changes in his metacognition through the course of treatment in the concluding section of the case vignette.

Element One: Agenda

When beginning therapy with Ryan, he would often bring a list of items to discuss in sessions. These were ideas that he collected throughout the week when things would happen at school, during interactions with peers, or at home. He expressed that he wanted to “do well” in therapy so that he could progress and get better. It quickly became apparent that these items were often things that Ryan already felt comfortable handling or that he already had coping strategies in place to try to address distress when it would be present. The therapist was able to share this observation with Ryan and offered the remark that “you aren’t graded for how well you do in therapy, it seems like you want to get an “A” in session”. This observation served to diffuse some of the tension that Ryan felt. He expressed that when he does new things, he wants to do them well, and the thought of “failing” as a therapy patient felt bad to him. This offered a space for the therapist to continue reflection with Ryan and offer that perhaps it was more comfortable thinking about things that Ryan already felt good at but that there was a chance there could be utility in sharing things he felt uncertain or confused about. Ryan expressed a fear of being uncomfortable or losing control. He often came to the conclusion that he was a bad person if he talked about the things that were on his mind but agreed that therapy was likely a good place to try discussing such things.

As treatment progressed with Ryan, he became more comfortable sharing his internal states and experiences with his therapist. He identified that one of the things that helped him to feel more comfortable sharing was the understanding that his time with the therapist was his own, that he had the freedom to explore topics, while also having his therapist’s guidance to weave a line through the topics shared back to the psychological problems he was experiencing.

Element Two: Insertion of Therapist’s Mind

A critical element of the therapist’s work with Ryan was the insertion of the therapist’s mind. Ryan expressed feeling relief when the therapist would share narratives from his own life that were related to a topic that Ryan would be exploring in session. For example, Ryan arrived at a session feeling panicked because he felt like he had “lost control” in front of a group of his friends when he had been feeling overwhelmed and frustrated. Ryan shared that he had not liked how a game was going, so he threw his game piece across the room, and shouted in frustration. He concluded that this was proof that he was filled with overwhelming rage and meant that he was an intrinsically bad person. The therapist took the opportunity to share a narrative from his own life when he felt like he had lost control in front of others and had felt embarrassed about it and concerned that people would think negatively of him as a result. The therapist asked Ryan what it was like to hear this narrative from the therapist’s life. Ryan remarked that it was relieving to hear this story, that it humanized the therapist, and that it was possible to have a full range of experiences without being deemed “all good or all bad”. These moments of self-disclosure from the therapist also served to help Ryan identify his emotional states, his reflections on his own thoughts and behaviors, and reflections on the complexity of Ryan himself and others in his life.

Element Three: Narrative Episode

At the outset of the sessions with Ryan, it was challenging for him to share narrative episodes from his life. There would be limited detail in the narratives, it would be confusing to ascertain who else may be involved, and Ryan often expressed feeling confused when trying to share a story from his life. The therapist observed that when Ryan would try to share narratives, he seemed to become increasingly disorganized and frustrated. The therapist also observed that many of the narratives were from the recent past (within a few days or weeks) and there were few narratives present from when Ryan was a child or early teen. One strategy that was helpful was for the therapist to assist in scaffolding the narratives by sharing stories from the therapist’s own life that matched the themes that Ryan was exploring.

As Ryan became more proficient at identifying his internal states, some of his narratives became more organized and richer in detail. Eventually, Ryan identified that it was challenging for him to think about stories from the past because they were distressing to think about; that the details felt confusing because he himself had felt confused in those moments; and that he had a concern that if he thought too much about them that he would not be able to move past that distress. These were helpful moments for Ryan and the therapist to think about ideas related to thought–action fusion, meaning making, resilience, and building mastery strategies that addressed Ryan’s psychological problems.

Element Four: Psychological Problem

When first beginning therapy, Ryan’s sense of his psychological problem was that he was “damaged goods”, “not a real person”, and “destined to hurt people” in his life. Most of Ryan’s thoughts were absolutist and he was certain that most of these issues would not resolve. Ryan expressed that the presence of symptoms was extremely distressing and that, unless they went away entirely, he felt like he would not have the life he desired.

Over the course of treatment, Ryan and the therapist were able to break these ideas down into more refined psychological problems by exploring narratives related to the ideas, thinking about their own therapeutic relationship, and participating in some family based interventions. It became apparent that Ryan held a tremendous amount of anger within him, that he was terrified that this anger meant he was a bad person, and that this anger made it challenging for him to meaningfully participate in his close relationships. Additionally, Ryan and the therapist identified that he felt a lot of insecurity in his relationships. Parts of him desired to be cared for, while other parts of him desired independence and were reactive to perceived limitations to his self-determination.

Ryan’s psychological problems also changed over the course of treatment as Ryan’s metacognitive capacities shifted and changed. Similarly, his abilities to identify what his psychological problems were also changed as he became more comfortable and familiar with his internal states. His belief that he was damaged and unable to change evolved into a more nuanced idea that he was not more flawed than other people and that his problems were understandable in the context of his life and could be improved upon.

Element Five: The Therapeutic Relationship

When first beginning therapy, Ryan shared that it was strange being open and honest with an adult. He shared that, with most of the adults in his life, he had a desire to be perceived as a “good kid” and did not want to share anything that would cast a negative light onto him. Therefore, talking about things that could be perceived as worrisome felt challenging. Ryan and the therapist needed to establish a sense of trust in their relationship so that Ryan felt comfortable exploring vulnerable topics. One of the methods that was most beneficial was thinking about the relationship out loud and reflecting on it together. The therapist would ask “what is it like to talk to me about these things?” “how are my reactions different than other adults in your life?” “do I remind you of anyone that is in your life?” “are there things that you wish could go differently/better when we talk and think together?”

An aspect of the therapeutic relationship that Ryan identified as particularly helpful and strong was the therapist’s open and curious stance about Ryan’s lived experiences, thoughts, and emotions. Ryan expressed that he felt like he could share things that often felt “bonkers” to him and the therapist would meet him with curiosity instead of judgement, and a sense that things can be rational and understandable even when they felt confusing and distressing.

Element Six: Progress

Throughout the course of treatment, Ryan and the therapist would often think about the element of progress. This was a crucial reflection in their work together because Ryan expressed at the beginning of treatment that he felt like he would have these psychological problems forever because they had already been going on for so long. For Ryan, it was often challenging to identify how he felt like he was changing or progressing in sessions. Therefore, it was critical that the therapist explicitly identified ways in which he perceived change in Ryan. This was sometimes a point of contention. Ryan would express frustration that others could perceive change in him when he could not detect it himself. Additionally, he expressed that some of the enhancements in his own metacognition served to increase his perceived distress. These moments served as opportunities to reflect on the meaning of recovery, meaning making, and resilience. A reflection that Ryan found most helpful and would return to often was the idea that wellness did not mean the absence of symptoms or distress but the understanding that he could succeed even when symptoms and distress were present.

Element Seven: Stimulating Self-Reflectivity and Awareness of Others’ Minds

At the beginning of the therapy services, Ryan had metacognitive deficits in both S and O. He was able to identify his internal states, but the identification lacked nuance and understanding. This was also consistent in the O domain, in that he was able to perceive internal states of others but was significantly confused about what they could mean or whether he was detecting them appropriately. For Ryan and the therapist, helping Ryan to first recognize his internal states at a physical level assisted in a more complex understanding of his emotions and thoughts over time. Asking how he felt an emotion in his body helped him to conceptualize when that emotion was present, and was helpful in exploring narratives about when he had felt that way recently and in the past. This was also helpful during therapy, as Ryan could identify when he started to feel anxious, angry, and sad while in session. Ryan shared that, when he started to detect these emotions in session, it meant he should talk about them with the therapist instead of tamping them down or ignoring them.

As Ryan’s self-reflectivity grew, he was able to form more complex ideas about himself, others, and his psychological problems. Where he once felt distress over the experience of a perceptual anomaly, he could now invite himself to be curious about the experience and identify a possible function of the symptom. An example he shared was “at some point I started to feel really isolated and lonely, so the presence of an “other” in my mind helped me feel like I had some company or someone I could talk to. But now I have people in my life, so I rather that voice just go away so it’s annoying that it hasn’t”.

Element Eight: Stimulating Mastery

When first beginning therapy, Ryan’s sense was that his problems were fixed and unchangeable. He demonstrated internalized stigma around having psychosis-like experiences and had the belief that something was intrinsically wrong with him, that he must be a bad person. Targeting these assumptions and beliefs in session were critical. To stimulate mastery, Ryan and the therapist began with identifying a plausible psychological problem in Ryan’s life, identifying how it was distressing to him, and obtaining a sense of how he felt he could navigate this problem. Over time, Ryan identified that he mostly felt overwhelmed by his problems and was unsure how or if he had been coping with them at all. By exploring narratives, enhancing his awareness of his internal states, and thinking about both practical interventions and increasing his sense of meaning making, Ryan was able to move from gross avoidance to a more nuanced approach to targeting his problems. As his metacognitive capacities increased, so too did his utilization of mastery strategies, particularly in his perceived sense of resilience.

Termination of Treatment and Outcomes

Ryan and the therapist met routinely over the course of four years, meeting weekly for one hour sessions for the first two years and then moving to biweekly over the course of the last two years. During this time, Ryan graduated high school, began working full-time, and then enrolled in a full-time undergraduate program at a local college. By the end of psychotherapy, Ryan had made significant improvements in his metacognitive abilities. At the time of termination, Ryan expressed that he felt like he was still growing and developing his understanding of himself and the world around him, and that he had made significant progress. His self-reflectivity capacities improved to the point where Ryan was able to recognize patterns of functioning over time and was now forming a complex personal narrative about his life that included his awareness of how his behavior was impacted by his thoughts and feelings. Similarly, Ryan’s awareness of others also grew to reflect his ability to recognize what others may be thinking and feeling and make reasonable hypotheses about the intentions of other people and what their cognitive and emotional functioning might be, based on verbal and nonverbal cues. In terms of decentration, Ryan, at the time of termination, had the ability to understand that events in life could be seen from multiple valid perspectives and that those perspectives may differ from his own beliefs and perspectives. Finally, Ryan significantly improved in the domain of metacognitive mastery. He was able to cope with his psychological problems, using a complex understanding of how his beliefs, perceptions, and thoughts interacted to contribute to the development of his psychological problems. Additionally, he demonstrated the ability to respond to his psychological problems by altering how he thought about them and formed adaptive responses to subjective distress. Taken together, his metacognition on the MAS-A scale at the end of therapy was as follows: self-reflectivity (8), understanding mind of the other (5), decentration (2), and mastery (7).

## 13. Discussion

Offering high-quality recovery-oriented interventions to patients experiencing CHR for psychosis is essential in helping them to make sense of their distress and to hopefully avoid prolonged suffering. In this paper, we have outlined MERIT (one such intervention that can be used with CHR patients) and shared key reflections from our work implementing MERIT with this population, including a case vignette of utilizing MERIT with a CHR patient. One conclusion that seems clear from our experience of implementing MERIT in a CHR clinic is that MERIT can be practiced to fidelity with a CHR population. The inherent flexibility of MERIT makes it an ideal treatment to use with this population and provides a framework from which a range of interventions may be offered to meet the unique needs of the CHR population in our clinic. There are key themes and considerations for MERIT therapists to keep in mind while offering MERIT to patients with CHR.

One theme that is mentioned across several elements is the challenge of promoting a non-hierarchical relationship, given the likely age and experience difference between patients and the therapist. Given that individuals with CHR are often youth, therapists are more than likely going to be older than their patients in a CHR clinic. The stage of life that youth tend to be in, and the age difference between them and their therapist, make it likely that patients with CHR will tend to view their therapist as an authority figure and expect the therapist to tell them what to do; in fact, many youth hope for this outcome, seeing it as a positive to have someone give them advice or tell them how to navigate their problems. In addition, for many of the patients seen in our clinic, this is their first encounter with the mental health system and, as such, there are more instances in which psychoeducation (regarding psychosis, recovery, stigma, and mastery strategies for distress) are warranted than is often the case in work with other (e.g., older) populations. As this might reinforce the notion that the therapist is an expert to whom the patient should defer, the therapist often must attend to and address these dynamics in both direct and indirect ways. Direct ways of balancing this conflict include naming it clearly in the session (i.e., “You want me to tell you what to do”) and then exploring that desire. This would support the patient’s autonomy by asking them to reflect on their own ideas about the situation and why they are seeking input from the therapist. Indirect ways of managing the balance between the therapist being more active and supporting the patient’s autonomy include noticing when the patient has instances in the session or their lives when they are autonomous and supporting reflection around these experiences, helping the patient to become more aware of their own desires and wishes.

As a treatment that is recovery-oriented and focused on helping patients to recover a sense of agency and sense of self by focusing on narratives, MERIT assists patients in developing self-direction and management of their lives. This is important to any population being offered psychotherapy but may have specific importance in helping youth with CHR to develop autonomy as they move into adulthood. The focus on developing metacognitive capacity can assist youth in having more integrated understandings of self and others that can help them manage life’s challenges as they mature.

Of note, there are limitations to the current paper to consider. One such limitation is that the material presented is reasoned from clinical experience. Further work is needed to more systematically study the efficacy of MERIT in CHR populations, including randomized controlled trials. Additionally, the conclusions presented here result from work with a limited CHR population and a small number of specific therapists; thus, it would be important to explore the use of MERIT with a larger and more diverse group of patients experiencing CHR and its utility with a wider range of therapists. There are also limitations of sample size, as the CHR clinic in which these findings were gathered serves a small number of patients. As such, in addition to more systematic trials of MERIT with CHR, future research should investigate the effectiveness of MERIT with a larger sample size and more diverse clinical population. The findings presented here may have limited generalizability to other settings and populations given the small sample size of both patients and therapists, and the possibility of therapist bias, as all therapists and supervisors are trained in MERIT. The field would benefit from further research to evaluate the efficacy of MERIT with a wider range of patients and therapists from more diverse training backgrounds. Further work is needed to assess whether MERIT offers improved outcomes compared with other treatment modalities, and to determine the long-term outcomes for individuals with CHR receiving MERIT.

## Data Availability

No new data were created or analyzed in this study. Data sharing is not applicable to this article.

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
