# Peer review of "A Recovery-Oriented Approach: Application of Metacognitive Reflection and Insight Therapy (MERIT) for Youth with Clinical High Risk (CHR) for Psychosis"

_behavsci, 2024, doi:10.3390/bs14040325_

Round 1

Reviewer 1 Report

Comments and Suggestions for Authors

 The writing is generally logical and coherent. When the author tried to analyze a notable point about applying a technique, the illustration was somehow vague. For example, the author repeatedly mention that due to the limited ability of CHR population to form sense of self, the therapists were supposed to assist the patient in gaining this ability, while the autonomy of patient should also be enhanced. Ideally, that will function well. However, the intervention of therapists can impede the autonomy, even though therapists do it with a good intention. Obviously, the author has noticed the potential conflict under this situation, but I recommend that the author explain how to fix this conflict and how to reach a balance between intervention and autonomy during therapy, so that the illustration will be more concrete, and the article will be more convincing.

Comments on the Quality of English Language

For Element Two: Insertion of the Therapist's Mind, there is phrase in the second paragraph, which is "Therapists were attuned to the dynamic of being placed in an authoritative role..." From my understanding, this paragraph argued that therapist should not comply with the patients' will to view the therapist as a leader. Therefore, the use of "attuned" here seems ambiguous to me. I hope the author can make this sentence clearer.

Author Response

 The writing is generally logical and coherent. When the author tried to analyze a notable point about applying a technique, the illustration was somehow vague. For example, the author repeatedly mention that due to the limited ability of CHR population to form sense of self, the therapists were supposed to assist the patient in gaining this ability, while the autonomy of patient should also be enhanced. Ideally, that will function well. However, the intervention of therapists can impede the autonomy, even though therapists do it with a good intention. Obviously, the author has noticed the potential conflict under this situation, but I recommend that the author explain how to fix this conflict and how to reach a balance between intervention and autonomy during therapy, so that the illustration will be more concrete, and the article will be more convincing.

We thank the reviewer for pointing out this interesting dilemma and have added the following text to address the potentially conflicting nature of promoting autonomy through intervention and offer more concrete examples:

Direct ways of balancing this conflict included naming it clearly in the session (i.e., “You want me to tell you what to do,”) and then exploring that desire. This would support the patient’s autonomy by asking them to reflect on their own ideas about the situation and why they were seeking input from the therapist. Indirect ways of managing the balance between the therapist being more active and supporting the patient’s autonomy included noticing when the patient had instances in the session or their lives in which they were autonomous and supporting reflection around these experiences, helping the patient to become more aware of their own desires and wishes.

Comments on the Quality of English Language

For Element Two: Insertion of the Therapist's Mind, there is phrase in the second paragraph, which is "Therapists were attuned to the dynamic of being placed in an authoritative role..." From my understanding, this paragraph argued that therapist should not comply with the patients' will to view the therapist as a leader. Therefore, the use of "attuned" here seems ambiguous to me. I hope the author can make this sentence clearer.

We have replaced the word “attuned” with the word “attentive” to help point out that therapists were attending to this dynamic in the therapy as it arose.

Reviewer 2 Report

Comments and Suggestions for Authors

This paper is interesting, easy to read, and essentially simple in its content and structure. It provides a description of the rationale and potential advantages of offering MERIT (an integrative group psychotherapy approach mainly focused on metacognition) to CHR individuals, mainly reasoned from preliminary clinical experience.

Neither original results or literature data are presented here, and the paper is substantially configured as theoretical. 

Most al all, I suggest an appropriate revision of English language in this manuscript (as mentioned below).

Then, I think that more information could be provided regarding the prevalence and impact of the CHR condition, and, most importantly, regarding potential outcomes and quantitative/qualitative estimates of recovery rates and dimensions for this population.  More can be also reported on other different psychological/psychosocial interventions that could be perceived as suitable for these individuals. Finally, specific evidence on MERIT is only mentioned, and is not described. I therefore suggest to expand the reference list and pertinent text with regard to all these aspects: this could be useful to increase the interest for the reader and the scientific relevance of this work.

Comments on the Quality of English Language

Although the paper content is sufficiently clear on the whole, extensive editing of English language could be useful: many grammatical errors and mistakes in sentence construction can be spotted throughout all the section of the manuscript. 

Author Response

This paper is interesting, easy to read, and essentially simple in its content and structure. It provides a description of the rationale and potential advantages of offering MERIT (an integrative group psychotherapy approach mainly focused on metacognition) to CHR individuals, mainly reasoned from preliminary clinical experience.

Neither original results or literature data are presented here, and the paper is substantially configured as theoretical. 

Most al all, I suggest an appropriate revision of English language in this manuscript (as mentioned below).

We thank the reviewer for attention to these details, we have proofread the document and corrected any grammatical errors we saw. We are all native English speakers and believe the manuscript is clear and using correct English. We would be happy to make additional edits as needed if the reviewer or editors can provide further input on where errors are in the manuscript.

Then, I think that more information could be provided regarding the prevalence and impact of the CHR condition, and, most importantly, regarding potential outcomes and quantitative/qualitative estimates of recovery rates and dimensions for this population.  More can be also reported on other different psychological/psychosocial interventions that could be perceived as suitable for these individuals. Finally, specific evidence on MERIT is only mentioned, and is not described. I therefore suggest to expand the reference list and pertinent text with regard to all these aspects: this could be useful to increase the interest for the reader and the scientific relevance of this work.

We have added the information about the prevalence, impact, outcomes, and treatments for CHR as requested:

The prevalence of CHR is still being investigated (APA, 2013) but recent research reported rates of 19% in clinical samples and a much lower rate of 1.7% in the general population (Salazar de Pablo et al., 2021). While it is uncertain which individuals experiencing CHR will go on to experience psychosis, research has shown that CHR individuals experience functional impairments, whether they experience a full psychotic episode or not (Salazar de Pablo et al., 2022). …).

A recent systematic review (Salazar et al., 2021) noted that most treatments for CHR include an emphasis on psychosocial programming, including family interventions, cognitive behavioral interventions, motivational interviewing, substance misuse interventions, and skills training.

We have expanded the sections on the evidence behind MERIT as a treatment:

“Empirical evidence for MERIT’s effectiveness has included two open trials of MERIT for individuals experiencing psychosis that both showed improved metacognition, high rates of acceptability, and no adverse effects (de Jong 2016; Bargenquast et al., 2014). Randomized controlled trials have also shown good outcomes, including high levels of feasibility and acceptance in real world settings, improved metacognition, improved insight, and no adverse effects (de Jong et al., 2019, Hasson-Ohayon et al., 2023; Vohs et al., 2018). Additional evidence for MERIT’s effectiveness and acceptability come from a myriad of case studies examining MERIT with individuals with a range of presenting problems (see Lysaker et al., 2020 for a summary of case studies). Case studies do not provide the same level of evidence as more systematic trials, but present in-depth, rich accounts of how MERIT has helped unique individuals to move toward personal recovery. Several case studies and one trial that focused on MERIT’s potential to promote insight have been in FEP clinics (Leonhardt et al., 2016; Leonhardt et al., 2017; Vohs et al., 2018). Vohs and colleagues (2018) reported that individuals experiencing FEP who received 6 months of MERIT demonstrated clinically and statistically significant improvement in insight when compared to FEP participants who received supportive therapy in the control group. Two case studies by Leonhardt and colleagues (2016 & 2018) report in-depth analysis of how MERIT helped two individuals experiencing FEP to achieve meaningful gains in their metacognition and personal recovery.”   

Comments on the Quality of English Language

Although the paper content is sufficiently clear on the whole, extensive editing of English language could be useful: many grammatical errors and mistakes in sentence construction can be spotted throughout all the section of the manuscript. 

We thank the reviewer for attention to these details, we have proofread the document and corrected any grammatical errors we saw. We are all native English speakers and believe the manuscript is clear and using correct English. We would be happy to make additional edits as needed if the reviewer or editors can provide further input on where errors are in the manuscript.

Reviewer 3 Report

Comments and Suggestions for Authors

Comments on the Quality of English Language

Author Response

The article discusses the implementation of the MERIT (Metacognitive Reflection and Insight Therapy) framework in a clinical setting for individuals experiencing Clinical High Risk (CHR) for psychosis. It outlines the eight core elements of MERIT and reflects on their application with CHR patients. These elements include establishing the agenda, incorporating the therapist's perspective, eliciting narrative episodes, defining psychological problems, nurturing the therapeutic relationship, reflecting on progress, stimulating self-reflection and awareness, and promoting psychological mastery.

While the article provides valuable insights and reflections based on clinical experience, here are some suggestions for improvement:

We thank the reviewer for their thoughtful and considerate review of the manuscript.

  1. some terms and concepts introduced, such as metacognition and psychological mastery, could be explained more clearly to enhance understanding.

We have added the following to more clearly essential concepts of metacognition:

“Metacognition includes a spectrum of activities ranging from discrete to synthetic and is comprised of four domains: self-reflectivity (understanding one’s own mind), understanding the mind of the other, decentration (ability to see oneself as part of a larger whole), and psychological mastery (being able to apply reflection about self and others to respond to the challenges one faces in life) (Lysaker & Dimaggio, 2014). An example of discrete metacognition would include the ability to identify different cognitive operations in one’s own mind, such as knowing that one is having a memory or a desire. Synthetic metacognition involves integration of different aspects of oneself, such as understanding how an event earlier in one’s life may create a specific emotional state later in life. Individuals experiencing psychotic disorders have been shown to have deficits in metacognition (see Lysaker et al., 2020 for a full review), at both early and later phases of illness (Vohs et al., 2015) and at a more severe rate than other mental health conditions (Lysaker et al., 2019). Deficits in metacognition are connected to a range of poorer outcomes, including increased negative symptoms, poorer work performance, decreased intrinsic motivation, and impaired self-recovery (Arnon-Ribenfeld et al, 2017; Lysaker et al 2019; Kukla et al, 2013). In light of these connections with functioning and outcomes, metacognition is an important target for treatment.”

  1. while the article includes examples of applying MERIT with CHR patients, providing more detailed case studies or examples would help illustrate the implementation of each element more effectively.

We have added a case vignette of utilizing MERIT with one CHR patient, so that all elements can be explored in one case conceptualization.

  1. the article briefly mentions limitations but could benefit from a more thorough discussion. This could include addressing potential biases, sample size limitations, and the generalizability of findings to other settings or populations.

We have added the following to the discussion of limitations of the ms:

“There are also limitations of sample size, as the CHR clinic in which these findings were gathered served a small number of patients. As such, in addition to more systematic trials of MERIT with CHR, future research should investigate the effectiveness of MERIT with a larger sample size and more diverse clinical population. The findings presented here may have limited generalizability to other settings and populations given the small sample size of both patients and therapists, as well as the possibility of therapist bias as all therapists and supervisors were trained in MERIT.

  1. while the article draws from clinical experience, incorporating empirical evidence, such as outcomes from randomized controlled trials or other research studies, would strengthen the article's credibility and support its claims about the effectiveness of MERIT with CHR populations.

We have added a more thorough discussion of the evidence for MERIT to the relevant section of the introduction:

“Empirical evidence for MERIT’s effectiveness has included two open trials of MERIT for individuals experiencing psychosis that both showed improved metacognition, high rates of acceptability, and no adverse effects (de Jong 2016; Bargenquast et al., 2014). Randomized controlled trials have also shown good outcomes, including high levels of feasibility and acceptance in real world settings, improved metacognition, improved insight, and no adverse effects (de Jong et al., 2019, Hasson-Ohayon et al., 2023; Vohs et al., 2018). Additional evidence for MERIT’s effectiveness and acceptability come from a myriad of case studies examining MERIT with individuals with a range of presenting problems (see Lysaker et al., 2020 for a summary of case studies). Case studies do not provide the same level of evidence as more systematic trials, but present in-depth, rich consideration of how MERIT has helped unique individuals to move toward personal recovery. Several case studies and one trial one trial that focused on MERIT’s potential to promote insight have been in FEP clinics (Leonhardt et al., 2016; Leonhardt et al., 2017; Vohs et al., 2018). Vohs and colleagues (2018) reported that individuals experiencing FEP who received 6 months of MERIT demonstrated clinically and statistically significant improvement in insight when compared to FEP participants who received supportive therapy in the control group. Two case studies by Leonhardt and colleagues (2016 & 2018) report in-depth analysis of how MERIT helped two individuals experiencing FEP to achieve meaningful gains in their metacognition and personal recovery.

  1. ethical considerations, such as patient confidentiality, informed consent, and power differentials in the therapeutic relationship, should be discussed more explicitly to ensure ethical practice when implementing MERIT with CHR patients.

We have added the following to the manuscript:

“MERIT should be conducted after the patient has been able to give their informed consent and standard measures should be taken to ensure ethical practice, including safeguarding confidentiality.”

We have also added more reflection on power dynamics within the manuscript in the discussion:

“Direct ways of balancing this conflict included naming it clearly in the session (i.e., “You want me to tell you what to do,”) and then exploring that desire. This would support the patient’s autonomy by asking them to reflect on their own ideas about the situation and why they were seeking input from the therapist. Indirect ways of managing the balance between the therapist being more active and supporting the patient’s autonomy included noticing when the patient had instances in the session or their lives in which they were autonomous and supporting reflection around these experiences, helping the patient to become more aware of their own desires and wishes.”

  1. providing suggestions for future research directions, such as exploring the long-term effectiveness of MERIT interventions or comparing MERIT with other treatment modalities, would enhance the article's contribution to the field.

We have added the following to the manuscript:

“The field would benefit from further research to evaluate the efficacy of MERIT with a wider range of patients and therapists from more diverse training backgrounds. Further work is needed to assess whether MERIT offers improved outcomes as compared to other treatment modalities, as well as to determine the long-term outcomes for individuals with CHR receiving MERIT.

  1. the article could benefit from improved clarity and organization.

We have proof read the document and have attempted to clarify and improve the organization of the manuscript.

Round 2

Reviewer 3 Report

Comments and Suggestions for Authors

The article is much improved now. I would recommend to keep the steps of MERIT in the 3 rd chapter and maybe put the elements in Italics, not assign a new chapter for each element. I recommend the article for publication.